# A stochastic model for simulating ribosome kinetics *in vivo*

**Eric Charles Dykeman**[¤]

Department of Mathematics, University of York, York, United Kingdom

¤ Current address: Department of Mathematics, University of York, York, United Kingdom
* eric.dykeman@york.ac.uk

**Data Availability Statement:** All relevant data are within the manuscript and its Supporting Information files.

**Funding:** The author received no specific funding for this work.

## Abstract

Computational modelling of *in vivo* protein synthesis is highly complicated, as it requires the simulation of ribosomal movement over the entire transcriptome, as well as consideration of the concentration effects from 40+ different types of tRNAs and numerous other protein factors. Here I report on the development of a stochastic model for protein translation that is capable of simulating the dynamical process of *in vivo* protein synthesis in a prokaryotic cell containing several thousand unique mRNA sequences, with explicit nucleotide information for each, and report on a number of biological predictions which are beyond the scope of existing models. In particular, I show that, when the complex network of concentration dependent interactions between elongation factors, tRNAs, ribosomes, and other factors required for protein synthesis are included in full detail, several biological phenomena, such as the increasing peptide elongation rate with bacterial growth rate, are predicted as emergent properties of the model. The stochastic model presented here demonstrates the importance of considering the translational process at this level of detail, and provides a platform to interrogate various aspects of translation that are difficult to study in more coarse-grained models.

## Author summary

Biological processes that occur in the cell, such as protein synthesis by the ribosome, are ideal examples of complex systems where the observed properties of the process depend on the interactions between the various components which make up the system. In the case of protein translation, the interplay between ribosomes, tRNAs, elongation, initiation and recycling factors with the transcriptome results in several complex behaviours, such as increasing protein chain elongation rate with increasing cellular growth rates. A key question is how this complex phenomenon emerges from the interactions of the individual components. Here I develop a general computational method which takes into account the complexity of the translational process and demonstrate how several biological phenomena of translation emerge naturally as a result of modelling translation from a more detailed view.

**Competing interests:** The author has declared that no competing interests exist.

# Introduction

In cells, ribosomes are responsible for translation of protein from messenger RNA (mRNA), which has been transcribed from the genomic DNA. At any given time, multiple mRNAs are being transcribed by the polymerase machinery in response to cellular stresses and feedback mechanisms. The full set of transcribed mRNAs in the cell represents its transcriptome, and translation by the ribosomal machinery gives rise to its proteome. The dynamics of the transcriptome, its response to cellular stresses, and its influences on protein synthesis, as well as the overall proteome, are important for understanding how disease processes arise at the cellular level. Although there are many processes that can influence the dynamics of the proteome, such as control of mRNA expression at the transcriptional level, it has been recognised that the overall kinetics of translation by the ribosomal machinery can exert additional control on protein levels in the cell. Because translation by ribosomes involves many individual kinetic steps and is influenced by the concentration of translational factors such as elongation and initiation factors, which must be created during ribosomal translation of the mRNAs encoding them, the dynamics of the translational process has complex interdependencies and feedback loops making the accurate modeling of the process challenging.

In essence, translation is governed by the physics of molecular interactions and diffusion processes, thus making it amenable to study using standard chemical kinetics and the mathematics of the chemical master equation. Ideally, when constructing a chemical master equation model of translation, one would like to account for as many of the individual biochemical reactions in the translational process as possible that affect the overall efficiency of protein production. Zur et al. [1] recently highlighted a list of seven features which they contend are required to be included in a translational model, at a minimum, to provide a comprehensive picture of translation *in vivo*. To date, development of kinetic models of the dynamics of ribosomes have mainly focused on either (1) stochastic based methods [2–4], or (2) statistical approaches which, roughly speaking, treat mRNAs as lattices with the ribosomes moving along the mRNA lattice points according to certain hopping probabilities [5–7]. The ribosomes can be modeled as occupying single lattice sites or as extended objects occupying multiple lattice sites [8].

Stochastic based methods have considered ribosomes moving along the mRNA in a stochastic manner, with different levels of detail taken into account. For example, Chu and von der Haar [3] have constructed a tool in Java which takes into account the movement of 200k ribosomes on 60k mRNAs with sequence information for each, with positional information on the mRNA encoded for each ribosome. Their model of translation in yeast [2] considers the complex positioning of all the ribosomes on the mRNAs along with relative tRNA abundances and the effects of codon bias and competition between the tRNAs for the A-aite of the ribosome. However, they do not consider concentration effects from elongation factors such as eEF1 eEF2 (Ef-Tu and Ef-Tu/Ef-Ts in bacterial systems), exchange of GDP for GTP on these of elongation factors, nor the concentration and competition effects between charged tRNAs for Ef-Tu and formation of TC, nor pre-mature termination on sense codons.

In the case of statistical based methods, these models are referred to as "totally asymmetric simple exclusion processes" or TASEP models. These types of models are well established in the literature and have been used to model aspects of the translational process since the 60's and 70's. A comprehensive review of TASEP models (or its cousin the ribosome flow model—RFM [9] which is a mean field approximation of the TASEP) and their current capabilites is not feasible here, but can be found, along with an account of various other computational models of translation, in [1, 10] with a discussion of the benefits and limitations of each. However, while RFM and TASEP models have been useful in considering some of the concentration dependencies on the translational process, such as from the inclusion of tRNAs or Ef-G

[11, 12]; or have included some steps of the initiation process [13], or considered TASEP models with a mixture of a few mRNAs [14], it has been recently stated that there are "currently no models that consider all the fundamental mRNA translational aspects at a cellular level" [1, 9].

In this work, I have followed what could be termed a "bottom-up" approach to the construction of a translational kinetics model to be used for simulation of translation in a cell, with the goal of fulfilling the seven features noted by Zur et al. [1]. This approach is based on two underlying principles: (1) to encode into the model the current biochemical knowledge on all of the ribosome reactions that occur at every stage (i.e. initiation, elongation, and termination), and (2) to minimally adjust experimentally measured kinetic parameters, when required, such that the results of the model recapitulate features of the translational process that have been either observed directly or deduced from experimental measurements *in vivo*. A similar approach has been recently attempted by Matsuura et al. [15], where they created an ordinary differential equation (ODE) model, solved in MATLAB, which simulates the synthesis of a tri-peptide from a single mRNA species containing three codons. Their ODE model contains a reported 968 reactions and 242 components, a large number considering that only synthesis of a tri-peptide from a single mRNA type was considered. This highlights the complexity of the number of states and reactions that need to be modelled when scaling up to an entire cell with thousands of different mRNAs and tens of thousands of ribosomes, and why models including all the biochemical reactions that can occur have been difficult to develop.

Although a "bottom-up" approach has been attempted by Matsuura *et al.*, there are a number of large differences between their model and the one I report here. First, the Matsuura model focuses on simulating protein synthesis in an *in vitro* reconstituted protein synthesis system, whereas I focus on modeling the translational process *in vivo*. Second, their approach only simulates the synthesis on a single mRNA type with a single ribosome occupying the mRNA at any one time. In contrast, I consider protein synthesis on thousands of mRNAs, each with an explicit and potentially unique nucleotide sequence that are allowed to be in a polysome state. Finally, although Matsuura *et al.* use experimentally measured kinetic rates, they have not, to my knowledge, checked for robustness of the rates against experimental observations of *in vitro* reconstituted protein translation systems. The last point is particularly important to note since many experimental groups have had conflicting accounts over the years of both the range of kinetic rates for certain reactions (most notably for GTP/GDP binding to Ef-G [16, 17]) and have also proposed different models for the kinetic events and their order of occurrence on the ribosome. Moreover, Indrisiunaite et al. [18] has noted that many of these rates can be sensitive to buffer conditions, making verification of the model by comparing with experimental observations critical. By testing the kinetic rates and reaction models that have been proposed by experimental groups for the various stages of translation, I have been able to confirm which of these result in a quasi-steady state of translation *in vivo* (please see supporting information for full discussion). Although the model I report here is not definitive and likely requires additional adjustments to the kinetic rates to better fit to experimental observations, particularly premature termination and stop codon read through rates (see supporting information), it does demonstrate that a whole cell model of translation, taking into account all known protein factors and biochemical steps, is computable in a reasonable time. Finally, by taking into account (1) the explicit nucleotide content of the full transcriptome, and (2) all of the translational factors (e.g. Efs, RFs and tRNAs) and how their concentrations influence the translational process, the model reveals how several translational properties emerge naturally from the model due to the complex network of the interacting components that are obscured in more coarse-grained models, or models which neglected, e.g., exchange of GDP for GTP on elongation factors [4].

## Results

### Computational model of ribosome kinetics in vivo

Fig 1 illustrates the various kinetic steps of ribosome kinetics *in vivo* that has been implemented in the stochastic model reported here. Highlighting the main features of the stochastic model, it contains $\approx 50$ kinetic parameters for the initiation, elongation, and termination stages of the ribosome, along with $\approx 20$ kinetic parameters which govern, for example, GTP/GDP binding to GTPases such as Ef-G and Ef-Tu. It is capable of simulating translation on thousands of unique mRNA sequences in a polysome arrangement, whose sequences are explicitly taken into account, and is set-up to predict initiation rates based on mRNA secondary structure and sequence at both cognate and non-cognate start codons (see supporting information). It also depends on the concentrations of: $\approx 10$ initiation, elongation, and recycling factors; nucleotide tri and di-phosphates; 20 amino acids; and over 40 tRNAs. Premature termination is accounted for in the model, along with mis-incorporation of non-cognate peptides, and the occurrence of stop codon read-through events. Although a framework has been set up for these advanced features in the code and the kinetic model for initiation and premature termination is based on the current best understanding of the biochemical steps involved, more experimental/theoretical work on kinetic rates would be required to implement these features in full. Moreover, as specific kinetic information for the aminoacylation of tRNAs have only been examined for two of the 20 aminoacyl-tRNA synthases, the complex individual kinetic steps are implemented in the model as a single reaction (dashed box in Fig 1A). It should also be noted that the model currently assumes that the concentrations of amino acids and nucleotide phosphates are constant, that mRNAs are not degraded or produced due to transcription, and that the *total* number of ribosomes, initiation factors, elongation factors etc. are constant. Thus, the model assumes a quasi-steady state for ribosome, mRNA, and elongation factor concentrations. This is a reasonable assumption for cells undergoing exponential growth, as a constant supply of amino acids and GTP/ATP would be expected in such a scenario. However, with minor modifications to the code, a more detailed model can be implemented where such features were dynamic. A full description of the model, along with justification for the kinetic rates based on experimental evidence, is given in the supporting information.

To simulate the kinetic reactions listed in the supporting information, I implement an exact Gillespie stochastic model [20] where individual reactions are randomly "fired", one at a time, according to the reaction propensity function

$$\Phi = \sum_{i=1}^{N}\sum_{j=1}^{M_i} k_{ij} = \sum_{i=1}^{N}\phi_i. \tag{1}$$

Here, $i$ labels one of the $N$ mRNAs, $j$ labels one of the $M_i$ possible reactions for this mRNA, and $k_{ij}$ are the individual kinetic rates for each reaction, with $\phi_i$ denoting the sum of reactions involving mRNA $i$ only. Since an exact Gillespie model can take considerable time to simulate, I am using a binary tree structure, with nodes representing the partial sums $\phi_i$ of the reactions involving mRNA $i$, to compute $\Phi$ and identify reactions to fire at each step. The tree is searched from the top node, choosing the branch which is greater than $r\Phi$, with random $r \in [0, 1]$, until a specific mRNA number $\eta$ which satisfies $\sum_{i=1}^{\eta} \phi_i > r\Phi$ is identified. After firing of a specific reaction in mRNA $\eta$, the reactions in this mRNA are updated and the same path through the tree is re-traced, updating the sums accordingly. In this way, the total propensity $\Phi$ is updated in $\log_2(N)$ time following the approach used in a Gillespie model of RNA kinetics involving single base-paring reactions [21]. This technique amounts to a re-ordering of the reaction

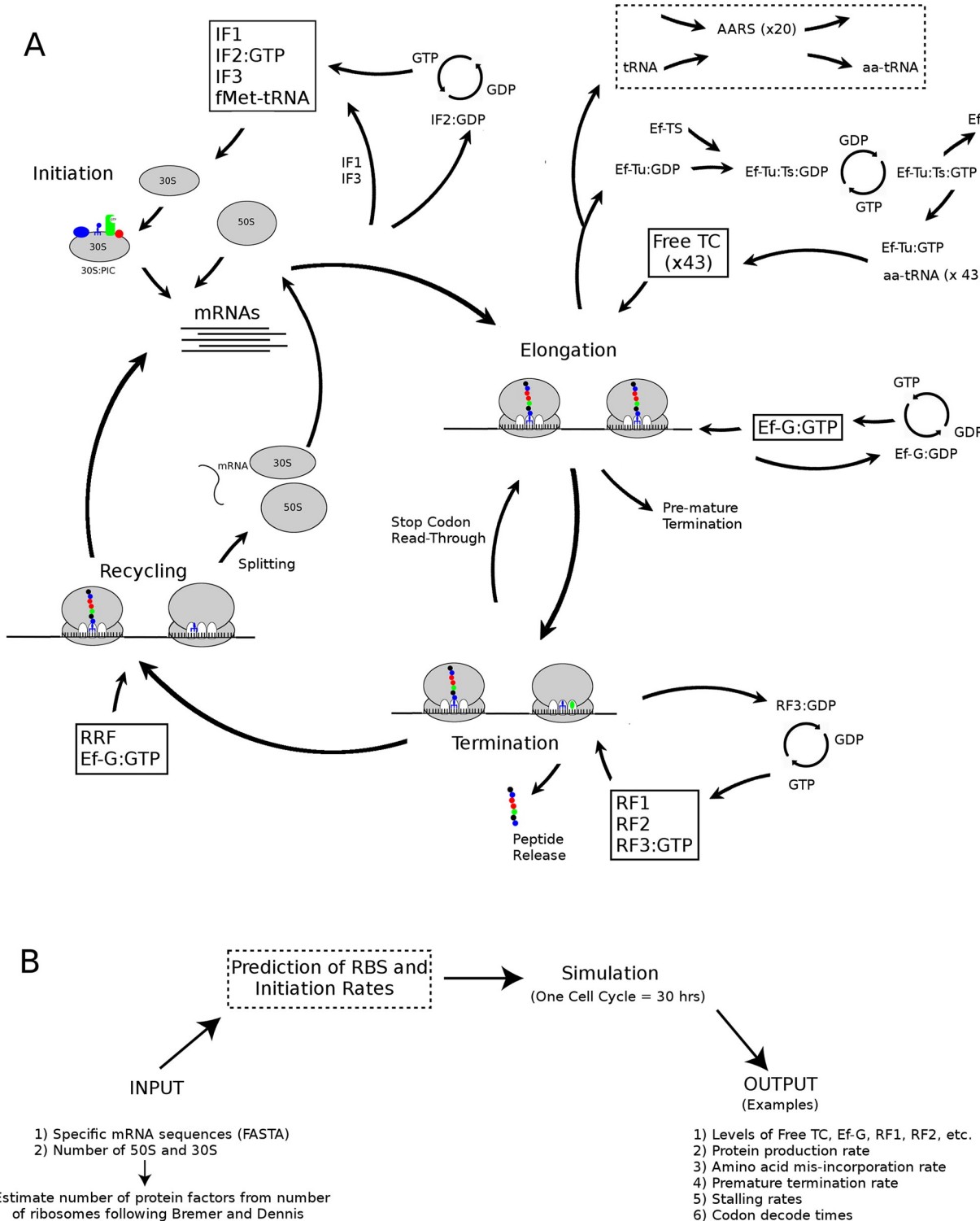

**Fig 1. Diagram of the ribosome kinetic model.** (A) Illustration of the reaction network that is simulated in the model. Aminoacylation of tRNAs (dashed boxes) indicate an area where there is limited knowledge of the biochemical kinetics and additional experimental work is needed. (B) Overview of the implementation steps. Specific mRNA sequences, and the concentration of ribosomes, are used as input parameters. The program determines the quantities of elongation factors, etc. that are expected from the number of ribosomes based on experimental data [19]. Examples of output data are shown in the final column.

propensities and, as noted by Cao et al. [22], results in a trajectory which will be statistically equivalent to one that was formed by the standard Gillepsie algorithm. Full details of the Log(N) binary tree method can be found in the supporting information of reference [21]. The simulations reported in this work for the $\mu = 1.0$ doublings/hour case take half a day to reach $10^3$ seconds of simulation time on a single processor. Simulations at the higher growth rate $\mu = 2.5$ take 3-4 days due to the larger number of ribosomes present and increased number of reactions that need to occur to reach $10^3$ seconds of simulation time. However, individual reactions are implemented for both cases in $\log_2(N)$ time. Thus, if a cell is doubling at a rate of once per hour ($\mu = 1.0$), then the kinetic model reported in this work can simulate the entire protein production that takes place in a single cell during one complete cell cycle in roughly 30 hours of computational time.

## Validation of the model with experimental observations

To verify that the model recapitulates experimental observations of translation in Prokaryotes, I have performed a simulation of translational kinetics in exponentially multiplying *E. coli* at three different growth rates; $\mu = 0.7$, $\mu = 1.0$, and $\mu = 2.5$ doublings per hour. Before simulating these, a representative transcriptome must be constructed that mimics what could potentially be observed *in vivo*. Each *E. coli* cell will have a temporaly dynamic transcriptome due to expression of different genes in response to various environmental cues and/or being at different stages of the cell cycle. Thus, an exact representation of the transcriptome of a single cell is difficult to measure, since measurments will often be averaged over many cells. Moreover, [23] have shown in a transcriptional time series measurement of the Lac operon using sm-FISH that the numbers of mRNAs for the LacZ gene not only vary in response to intracellular lactose concentration, but vary between cells following a geometric/Poisson distribution at low/high gene expression levels.

Using these observations, I have constructed transcriptomes by selecting genes from the *E. coli* proteome at random and assigning a number of mRNA copies for each gene sampled according to the two types of probability distributions (geometric/Poisson). It should be noted that the nucleotide sequence of the mRNAs do not match authentic *E. coli* mRNAs, but instead codons are chosen so that the codon biases match with the tRNA abundances in Table K in S1 Text (see Methods). This resulted in transcriptomes T07, T10, and T25 for use in the $\mu = 0.7$, 1.0, and 2.5 growth rate simulations, respectively. Although the T07,T10 and T25 transcriptomes are artificial, I will show in the next section (Biological Predictions of the model) that without the adjustment to the codon bias of the transcriptome to match tRNA abundances (or vis versa), translation efficiency is extremely poor.

The T07 transcriptome contains 650 mRNAs comprising $N_{nt} = 0.56$M nucleotides, while the T10 and T25 transcriptomes contain a total of 1265 mRNAs with 1.2M nucleotides and 5151 mRNAs with 5.0M nucleotides (c.f. Table 1). The total nucleotides in each transcriptome have been chosen to be roughly in line with experimental values from [19], which essentially fixes, to within approximately 10%, the total number of mRNAs that have been used in the simulations. In more concrete terms, the number of mRNAs that are used for the T07, T10, and T25 transcritpomes are within 10% of the value determined experimentally. Simulations were performed using the total number of elongation factors, tRNAs, etc. listed in Tables K-N in S1 Text. Each of these factors are assumed to increase linearly with the number of ribosomes in the cell following Bremer and Dennis [19]. Thus, users only need to specify the number of ribosomes and the individual mRNA sequences for the simulation (see Fig 1B). Kinetic rates were identical for all growth rates and are given in the supporting information. The volume of an *E. coli* ranges from $v = 0.83$ to $v = 1.12$ $\mu m^3$ for rates of $\mu = 1.3$ to $\mu = 2.1$ [19]. As this change

**Table 1. Efficiency of translation for different transcriptomes.**

| Trans. | N | $N_{nt}$ | $C_p$ | % I | % E | % T | % S | % 50S |
|---|---|---|---|---|---|---|---|---|
| T25 | 5151 | 5.00 | 21.0 | 5.1 | 77.4 | 2.0 | 2.3 | 15.6 |
| T10 | 1265 | 1.20 | 18.1 | 4.8 | 79.0 | 2.0 | 3.0 | 14.2 |
| T07 | 650 | 0.56 | 15.0 | 4.9 | 78.4 | 2.1 | 4.3 | 14.6 |
| T10a | 1265 | 1.20 | 8.1 | 3.8 | 95.0 | 1.0 | 13.6 | 1.5 |
| T10b | 2040 | 1.24 | 17.7 | 7.9 | 81.7 | 3.4 | 5.0 | 7.9 |
| T10c | 2040 | 2.00 | 17.9 | 6.6 | 90.0 | 2.1 | 1.0 | 2.3 |

Values for each transcriptome (T25,T10,etc.) are obtained from an average over three separate simulations. Variables denote the following quantities: $N$—number of mRNAs in the transcriptome, $N_{nt}$—total number of nucleotides in the transcriptome in millions, $C_p$—average protein chain elongation rate (aa/sec), (%I,%E,%T, and % S)—Percentage of ribosomes (I)nitiating, (E)longating, (T)erminating, (S)talled, and %50S—Percentage of 50S ribosomal subunits that are free.

in volume has negligible effect on the overall kinetic rates in a Gillepsie model, I have used an average volume size of $v = 1.0\ \mu m^3$ for all simulations. Finally, although my model can theoretically compute the ribosome binding site (RBS) and the initiation rate for an mRNA based on detailed knowledge of its 5' UTR sequence and secondary structure, this feature requires additional experimental validation before it can be fully implemented. Thus, I have assumed for now that ribosomes initiate on all mRNAs at the maximal rate, i.e. the situation where the 5' UTR is unstructured and a Shine-Dalgarno sequence is present. This has an added benefit for the testing/validation of the termination and recycling phases, which need to be fast enough to process mRNAs which are maximally translated. This requires that the kinetics of the termination and recycling steps be such that substantial stalling or a full blown traffic jam does not result on a maximally translating mRNA. Fixing mRNAs to the case of maximal initiation has allowed for verification that my model does not result in in-efficient translation on maximally translated mRNAs and has revealed that one particular experimental model of termination was more consistent with this requirement than an alternative (see supporting information).

Fig 2 shows the percentage of ribosomes at different stages of the translation process, along with the percentage of free 30S pre-initiation complex (30S:PIC) and free 50S, at the three growth rates. The percentages are computed relative to the total 50S (or equivalently 30S) subunits available. For all growth rates, roughly 15% of 50S ribosome subunits are free, while the remaining $\beta_r = 85\%$ are in one of the stages of translation. Specific values for intatition,

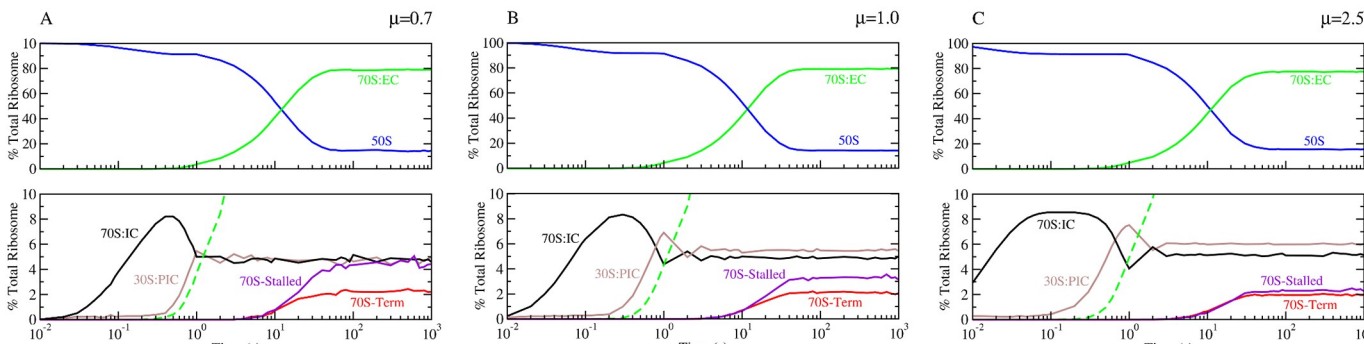

**Fig 2. Translational kinetics of ribosomes at different _E. coli_ growth rates.** Time courses for the percentage of ribosomes (compared with total ribosomal mass) that are initiating (black), elongating (green), terminating (red), stalled (purple), or free 50S (blue) and free 30S:PIC (brown) are shown for growth rates of (A) $\mu$ = 0.7, (B) $\mu$ = 1.0, and (C) $\mu$ = 2.5 doublings per hour. The experimentally expected ratio of ribosome in elongating complexes (70S:EC—green line) to total ribosome is 0.80–0.85 for all growth rates [19], which matches with the model prediction. Time courses represent an average of three separate simulations.

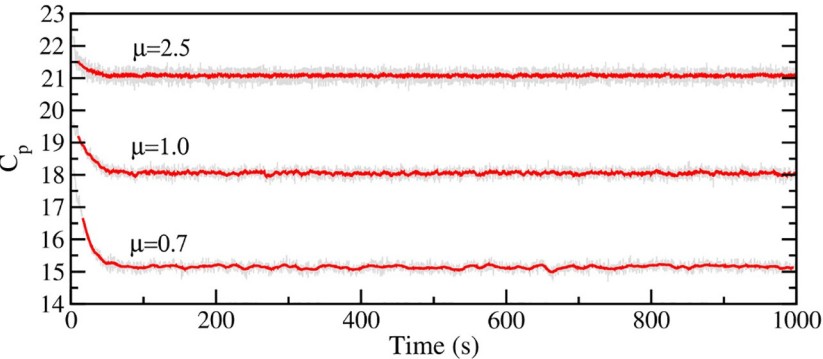

**Fig 3. Peptide chain elongation rates at different *E. coli* growth rates.** The peptide chain elongation rates ($C_p$) in amino acids per second are shown for the simulations at growth rates $\mu = 0.7, 1.0$, and 2.5 doublings per hour. Running averages of $C_p$ over 1000 seperate protein syntheisis events are given by the red curves and reveal an average peptide chain elongation of $C_p = 15, 18$, and 21 amino acids per second for $\mu = 0.7, 1.0$, and 2.5 doublings per hour, respectively. All model predictions of the peptide elongation rates match experimental estimates of [24].

elongation, and terminaition are listed in Table 1. The percentage of elongating ribosomes, which are stalled, is given in the %S column. Although the $\beta_r$ value is consistent with Bremer and Dennis [19], I have found that it has a complex dependence on at least four factors: (1) the total number of initiation sites, (2) the total number of nucleotides across the transcriptome, (3) the different lengths of the open reading frames and their relative abundance in the transcriptome, and (4) the initiation rates of the ribosome on all mRNAs. Finally, Fig 3 shows the computed peptide chain elongation rates ($C_p$) for all growth rates. As can be seen in Fig 3, the average $C_p$ value is roughly 15 codons per second at the growth rate $\mu = 0.7$, 18 codons per second at 1.0, while it is 21 codons per second at $\mu = 2.5$. These values compare extremely well with the expected rates of 15,18 and 22 codons per second, respectively, reported in Bremer and Dennis [19, 24].

## Biological predictions of the model

**Elongation rate increases with cellular growth rate.**   As can be seen in Fig 3, $C_p$ decreases as the growth rate decreases, even though all simulations use exactly the same kinetic rates for the elongation reactions. Previous models such as in Rudorf [25] reported on the use of different kinetic rates for the elongation phase of the ribosome in order to enforce the observed $C_p$ values, while the stochastic framework used in Vieira et al. [4] found that there was no increase in $C_p$ with increasing growth rates. Here, the difference in peptide chain elongation rates arises from the complex interdependence of the various translational factors, i.e it is an emergent property of the model. One might hypothesize that the decrease in $C_p$ that follows a reduction in the growth rate arises solely from the drop in concentration of the tRNAs and elongation factors, as these are directly associated with the elongation kinetics of the ribosome. This is incorrect, however, as the $C_p$ value, as well as the overall translational kinetics of the ribosome, are robust to attempts of increasing the protein chain elongation rate by optimizing the concentrations of a few translational factors. To demonstrate this, I have performed simulations at the growth rate of $\mu = 0.7$ doublings per hour with the same number of ribosomes and same transcriptome as the *T07* simulations shown in Fig 2A, but have doubled the number of certain components as follows. When *only* tRNAs and the elongation factors Ef-Tu, Ef-Ts, and Ef-G are doubled, $C_p = 15.1$, but with the additional consequence that there were twice as many 70S elongating ribosomes stalled on mRNAs (due to insufficient recycling factors). Doubling only

the number of Ef-Tu, Ef-Ts, or recycling factors resulted in values of $C_p$ = 16.0, 15.5, and 15.6, respectively. Finally, choosing different random transcriptomes (by sampling different genes) also had no effect on $C_p$. Only when all tRNAs, EFs and RFs doubled in unison did $C_p$ increase demonstrating how the peptide chain elongation rate is an emergent property of the model.

**Elongation rate depends on tRNA abundance and codon bias of the transcriptome.** Although the translational kinetics of the ribosome are robust to alterations of the translational factors, it is very sensitive to the abundance of tRNAs relative to the codon bias in the transcriptome. Dong *et al.* have previously measured tRNA molar abundances from *E. coli* at different growth rates using two-dimensional electrophoresis (c.f. Table 5 in [26]). Additonally, they estimated codon usage in the transcriptome at different growth rates using coding sequences from Genbank and previous measurements from [27] on the abundance of proteins at different growth rates (c.f. Table 3 in [26]). Using the Genbank and Pedersen data, Dong gives a codon usage frequency for AAA as 4.9% at $\mu = 1.0$. However, the tRNA cognate to AAA represents only 1.5% of the total molar mass of tRNA that they measured. Hence the codon usage reported by Dong, based on the Pederson and Genbank data, does not exactly match the tRNA profile that they measured. To illustrate the dramatic effects that this mismatching between tRNA abundance and codon usage in the transcriptome can have on translation efficiency, I have constructed transcriptome T10a, which was adapted from T10 so as to have identical numbers of mRNAs with identical reading frame lengths for each. However, the codons in T10a have been altered so as to have the same codon biases as the Pederson and Genbank data reported in Table 3 of [26] i.e. T10a has a codon bias reflecting the wild-type E. coli transcriptome. However, while the codon bias in transcriptome T10a reflects what was observed in the experiments of Pedersen et al. [27] for wild-type E. coli, the demand on tRNAs that this codon usage implies *will not* match the relative tRNA abundances from Dong et al. used in the simulation.

Simulation of translation using T10a following the same procedure as T10 reveals that elongation is substantially slowed, with an average elongation rate of $C_p$ = 8.1 codons per second when averaged over the entire transcriptome and a substantial 13% of ribosomes stalled (c.f. Table 1). As can be seen in Fig 4, the resulting ratios of tRNAs in free ternary complex (TC) for the T10a simulation varies across tRNA species much more then the T10 simulation, with some tRNA species only having 10% of their total number in free TC. This mismatch between the tRNA abundance and the codon bias of the transcriptome results in several tRNA species having very few of their total numbers available in free TC, resulting in increased decoding times at codons dependent on these tRNAs. However, in contrast with transcriptomes T07, T10, and T25, which have codon biases which match tRNA abundances, the percentage of tRNAs in free TC is roughly uniform across all tRNA species illustrating how the quantities of tRNAs match their rate of usage by translating ribosomes in these cases.

The results for the T10a transcriptomes are similar to a previous Markov model study by Rudorf et al. [28] which showed that tRNA abundances in free TC become skewed, relative to the total tRNA concentration, as a result of mismatches between the tRNAs and the frequency of codon usage in the mRNAs. Table 2 compares the ratio of tRNAs in free TC to the total amount of that tRNA for the transcriptome T10a (green data in Fig 3) with that of Rudorf et al. [28]. As can be seen from the table, there are similar effects on the tRNA abundances between T10a and those reported in Rudorf et al. [28]—e.g. the largest tRNA abundances are in tRNAs-Arg2,Arg3,Arg4, and tRNA-Arg5 while the least is in tRNA-Lys. However, there are some differences, and these are likely due to two factors. First in Rudorf and Lipowsky's model, Ef-Tu does not have a reaction for GTP/GDP mediated exchange by Ef-Ts, where as in my model this is present (c.f. Fig 1A). This is probably the largest contribution to the difference as roughly 21% of Ef-Tu is in complex with GDP or

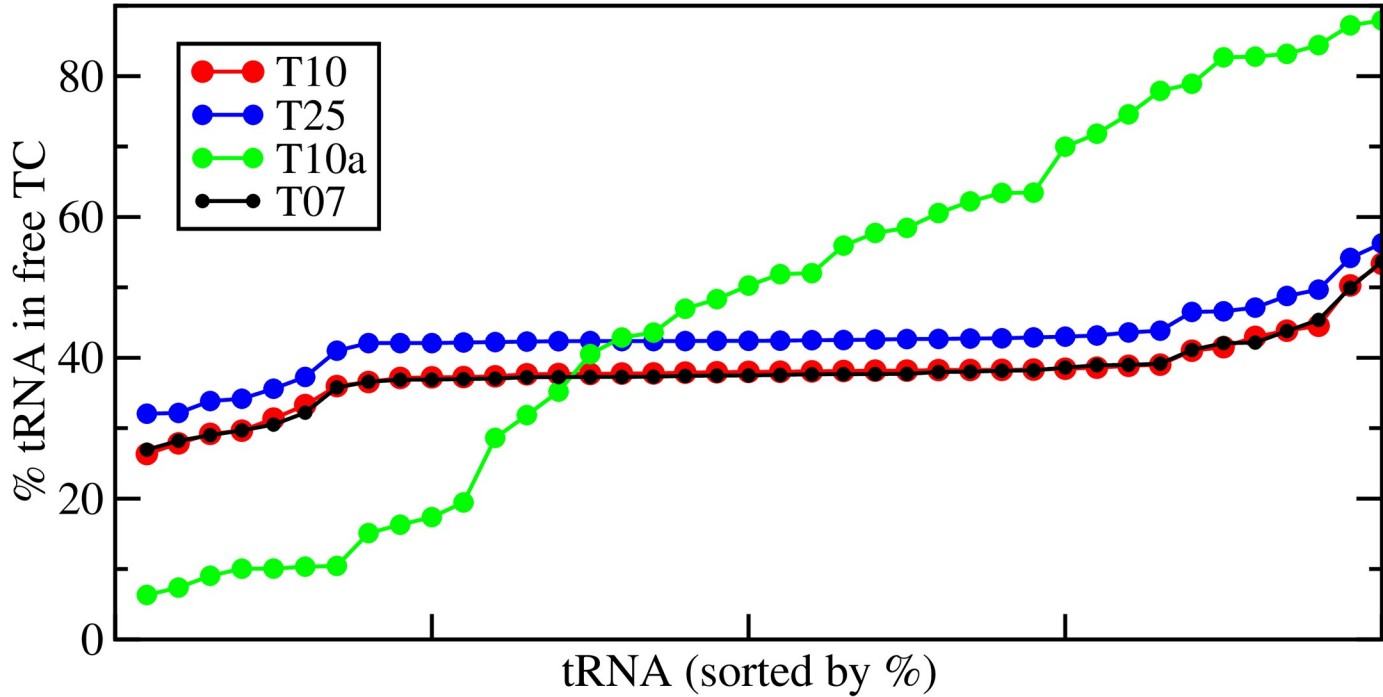

**Fig 4. Percentage of each tRNA in free ternary complex.** For each tRNA listed in Table K in S1 Text, the percentage of the tRNA that is in free ternary complex (TC) is computed by taking the ratio of the amount of the tRNA in free TC to the total amount of the tRNA. The average ratio for each tRNA (over the last 100 seconds of the translational simulation) is shown for for the T07 (black), T10 (red), and T25 (blue and T10a (green) transcriptomes. The tRNAs are ordered from lowest percentage to highest for each simulation separately.

Ef-Ts (at $\mu = 1.0$) in my model, and therefore unable to bind charged tRNAs to form free TC. Second, the effects of stalling are accounted for in this work, which can slow the time to release tRNAs which decode rare codons, thereby further decreasing their abundance in free TC. This effect can be illustrated in Fig 5 which shows my models predicted average decode times for each codon (in aa/sec), along with stalling frequencies, for each of the 61 sense codons. As can be seen, small effects of codon biases can dramatically alter the codon decoding times producing a feedback on the abundance of tRNAs in free TC, effecting the overall translation properties of the transcriptome.

The source of the slow average translation of the T10a transcriptome is due to the additional reactions governing GTP/GDP exchange on Ef-Tu and the binding of aminocyl-tRNAs to Ef-Tu:GTP to form free TC. These reactions create a situation where aminoacyl-tRNAs must compete for available Ef-Tu:GTP to form free TC. Underused aminoacyl-tRNAs will have an advantage in formation of TC over that of overused ones, resulting in a surplus of underused tRNAs in free TC. Thus, situations where the relative abundances of tRNAs do not match their demand implied by the codon bias of the transcriptome will result in shifts in the amount of tRNA available in free TC (as can be seen in Fig 4 for T10a). To further demonstrate the dramatic difference in translational behaviour that can be observed when all reactions are accounted for, I have performed a simulation where reactions governing the binding of Ef-Tu to newly charged tRNAs are turned off and tRNAs are imediately returned to free TC following ejection from an elongating ribosome. This is the same assumption used by Vieria et al. [4] and assumption 3 in Zouridis et al. [29]. Neglecting these reactions results in average translation rates of $C_p = 18.4$ aa/sec for T10 and $C_p = 17.7$ aa/sec for T10a, in contrast to the less efficient $C_p = 8.1$ aa/sec average translation that results for transcriptome T10a when these

**Table 2. Concentration of tRNA$^x$ in free ternary complex.**

| x | T10 | T10A | Rudorf |
|---|---|---|---|
| Ala1B | 0.3812 | 0.2862 | 0.5611 |
| Ala2 | 0.3776 | 0.1945 | 0.4935 |
| Arg2 | 0.3769 | 0.5187 | 0.6801 |
| Arg3 | 0.3713 | 0.8276 | 0.8551 |
| Arg4 | 0.3826 | 0.8439 | 0.8863 |
| Arg5 | 0.3832 | 0.8791 | 0.9006 |
| Asn | 0.3737 | 0.1043 | 0.4229 |
| Asp1 | 0.3789 | 0.3186 | 0.5356 |
| Cys | 0.3825 | 0.7182 | 0.8068 |
| Gln1 | 0.3726 | 0.6055 | 0.7520 |
| Gln2 | 0.3858 | 0.1511 | 0.4409 |
| Glu2 | 0.3765 | 0.4833 | 0.6505 |
| Gly1 | 0.4385 | 0.8315 | 0.8806 |
| Gly2 | 0.3135 | 0.7791 | 0.8512 |
| Gly3 | 0.3722 | 0.4287 | 0.6423 |
| His | 0.3844 | 0.1006 | 0.4258 |
| Ile1 | 0.3771 | 0.1003 | 0.6288 |
| Ile2 | 0.3799 | 0.8720 | 0.8253 |
| Leu1 | 0.3901 | 0.5593 | 0.7177 |
| Leu2 | 0.3818 | 0.6342 | 0.7524 |
| Leu3 | 0.2783 | 0.5201 | 0.6870 |
| Leu4 | 0.4099 | 0.8267 | 0.8680 |
| Leu5 | 0.2920 | 0.6998 | 0.8058 |
| Lys | 0.3778 | 0.0631 | 0.4149 |
| Metm | 0.3885 | 0.0903 | 0.4742 |
| Phe | 0.3655 | 0.0735 | 0.4731 |
| Pro1 | 0.4454 | 0.1629 | 0.5897 |
| Pro2 | 0.3807 | 0.6345 | 0.7566 |
| Pro3 | 0.2633 | 0.1034 | 0.4504 |
| Ser1 | 0.3327 | 0.5772 | 0.7239 |
| Ser2 | 0.5335 | 0.7890 | 0.8632 |
| Ser3 | 0.3801 | 0.5844 | 0.7240 |
| Ser5 | 0.4295 | 0.3518 | 0.5749 |
| Thr1 | 0.3818 | 0.1736 | 0.4444 |
| Thr2 | 0.5028 | 0.7457 | 0.8199 |
| Thr3 | 0.3818 | 0.1736 | 0.4470 |
| Thr4 | 0.2964 | 0.4357 | 0.6307 |
| Trp | 0.3829 | 0.6220 | 0.7641 |
| Tyr1 | 0.3794 | 0.5025 | 0.6666 |
| Tyr2 | 0.3794 | 0.5025 | 0.6666 |
| Val1 | 0.3597 | 0.4052 | 0.5835 |
| Val2 | 0.4151 | 0.4695 | 0.6494 |

The ratio of the concentration of tRNA$^x$ in free ternary complex relative to the total concentration of tRNA$^x$ at growth rate $\mu = 1.0$ is shown. Data for Rudorf computed from the 2-1-2 model at growth rate $\mu = 1.07$ in supplementary tables S4 and S5 [28].

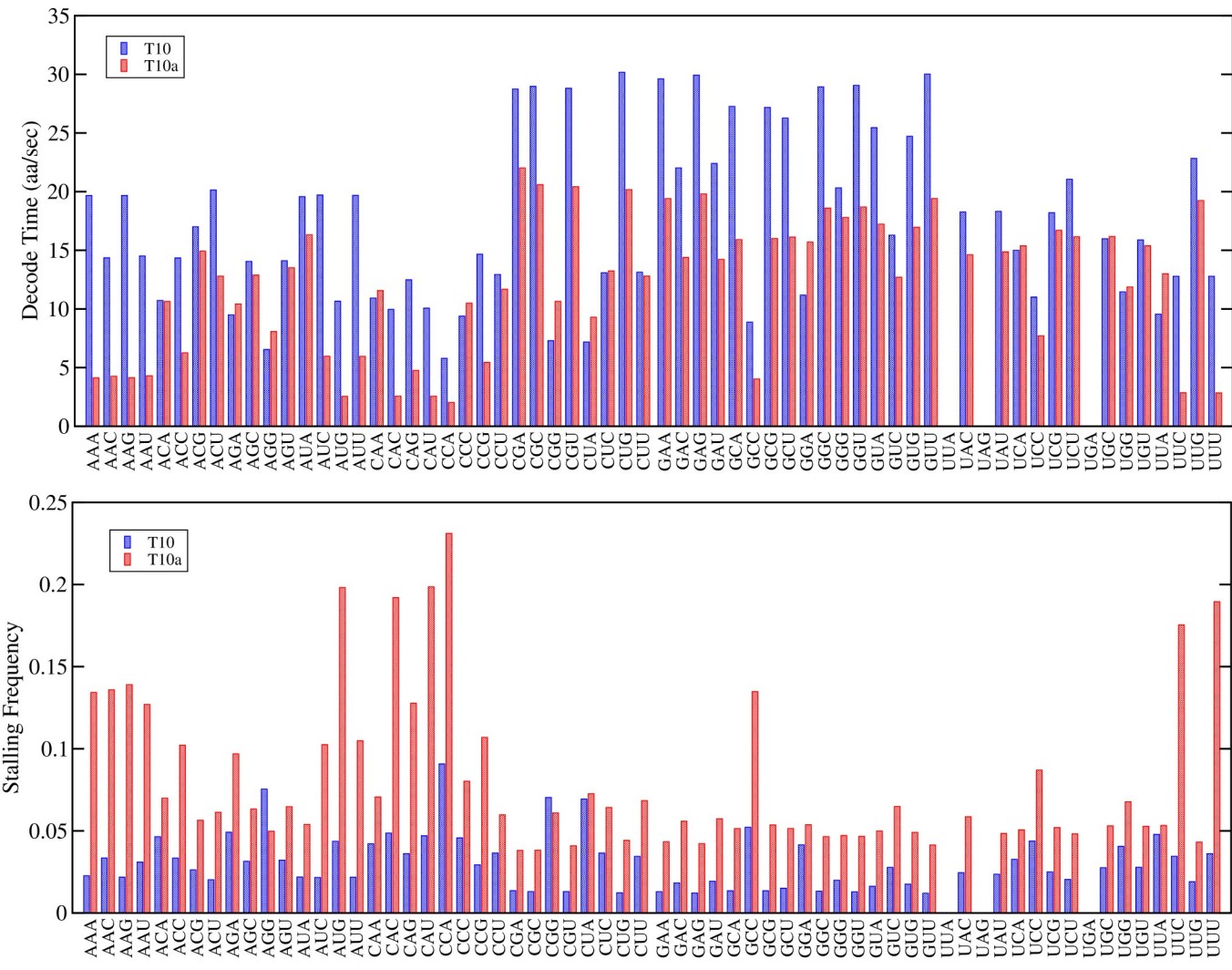

**Fig 5. Predicted codon decoding times and stalling frequency.** The average codon decoding times a stalling frequencies for each sense codon is calculated from the 215M individual decoding events that occur over the entire transcriptome in 1000s of simulation time. Blue and red bars indicate the T10 and T10a transcriptomes, respectively. Stalling events are computed as the frequency of ribosome stalling that occurs at the given codon.

reactions are accounted for. Similarly, increasing Ef-Tu (and Ef-Ts) concentrations by 25%, 50% and 75% results in $C_p$ values for T10a of 12.1, 14.2 and 15.1, respectively, reducing the effects of competition amongst the charged tRNAs for Ef-Tu:GTP as it becomes increasingly abundant. Both of these results are in close agreement with recent models of Rudorf et al. [28, 30] which have also taken into account the effects of competition amongst newly charged tRNAs for Ef-Tu:GTP and the effects of codon bias on the overall translational process.

**Ribosomal density on mRNAs is transcriptome dependent.** While the tRNA abundance relative to codon usage in the mRNAs appears to have a strong effect on translational efficiency, the length distribution of the mRNAs and their overall quantity control the average ribosomal density over the whole transcriptome (assuming the case of a fixed amount of ribosomal subunits). To illustrate this phenomenon, I have constructed two additional transcriptomes, T10b and T10c. The former, T10b, has essentially the same nucleotide content as T10 (1.24M nt in total), but is biased to short mRNA lengths between approximately 600 and 900

nucleotides. This results in transcriptome T10b, which has roughly the same total number of nucleotides as T10, but a higher number of overall mRNAs at 2040. The latter, T10c, has the same number of mRNAs as T10b, but has no bias in the length selection, and thus has a similar distribution of lengths as T10. Hence, T10c has both higher total nucleotide content (2.0M in this case) and more mRNAs overall compared with T10. To calculate the ribosomal density, I first calculate the average nucleotide distance between ribosomes following [19]

$$d_r = \frac{N_{nt}}{\beta_r N_{50}}, \tag{2}$$

where $N_{nt}$ is the total number of nucleotides in the transcriptome, $\beta_r$ is the ribosome activity, and $N_{50}$ is the total number of 50S ribosome subunits available. A density value is computed from $\rho = 1/d_r$, which is equal to the number of ribosomes per nucleotide. Note that $d_r$ has a minimal value, corresponding to the ribosomal footprint, indicating the maximal density of ribosomes that is physically possible. It should be noted that estimates for the ribosomal footprint vary from 10 to 14 codons [1, 31]. My model allocates a defined extended footprint (8 codons 5' of the P site and 6 codons 3' of it) for the ribosome in contact in the mRNA, following RNAase protection assays discussed in Berkhout et al. [31].

Table 1 shows the results for T10b and T10c using the same settings for rates and numbers of tRNAs, etc., as T10. Because initiation occurs at the maximal rate on all mRNAs and a consistent spacing between the first and second ribosomes is achieved on average in the simulations, one might assume that the average number of nucleotides per ribosome should only depend on the total nucleotide content, as assumed in [19]. However, as Table 1 shows, this is not the case as the overall center-center ribosome distance for T10b is $d_r = 86.5$, while for T10 it is $d_r = 92.9$, indicating a higher density of ribosomes in the T10b transcriptome. The increase in density results in more ribosomes being in a stalled state, with 5% of ribosomes now stalled compared with 3% in T10. Very interestingly, this increased stalling has only a tiny impact on the overall average elongation rate, with $C_p = 17.7$ for T10b, compared with $C_p = 18.1$ for T10. Thus, although there are more ribosomes stalled at any one time on the mRNAs, these must occur randomly and transiently so that any stalled ribosomes quickly begin moving again before causing additional ribosomes to stall behind them, building into a full traffic jam. Finally, simulating the transcriptome T10c, which has an increase in both the number of mRNAs and total number of nucleotides when compared with T10, it can be observed that the average distance between ribosomes is approximately $d_r = 135.9$, indicating a smaller ribosome density in T10c even though the ribosome activity is near 98%.

**Simulations using *E. coli* mRNAs.** Although the transcriptomes T07,T10, and T25 all reproduce expected features of the translational processes *in vivo* that have been highligetd in Bremer and Dennis [19], they are artificial transcritpomes which have codon biases that are not what would be observed in *E. Coli*. Thus, the question naturally arises as to the ability of the model to reproduce translational dynamics on real mRNA/transcriptome information. To probe this question, I have created transcriptomes T07wt, T10wt and T25wt which have near identical nucleotide and mRNA numbers as the artificial T07,T10, and T25 transcriptomes, but the mRNAs in these transcriptomes have been taken from the predicted open reading frames (ORFs) of *E. Coli* K12 strain MG1655 (accession code U00096). Table 3 reports on the average translational elongation rates using the tRNA abundances measured by Dong et al. [26], and compares with simulations using the alternative tRNA abundances shown in Table 4. The tRNA abundances in Table 4 represent one possible theoretical solution to the tRNA/ codon usage matching problem, i.e. the identification of the relative tRNA abundances such that the percentage of each tRNA out of the total matches the usage frequency of the codon(s)

**Table 3. Efficiency of translation for wild-type E. Coli transcriptomes.**

| Trans. | tRNA | $N$ | $N_{nt}$ | $C_p$ | % I | % E | % T | % S | % 50S |
|--------|------|-----|-----|-----|-----|-----|-----|-----|-----|
| T25wt | Dong | 5154 | 5.01 | 6.0 | 2.1 | 97.2 | 0.6 | 33.3 | 0.6 |
| T10wt | Dong | 1259 | 1.20 | 4.8 | 2.3 | 96.9 | 0.6 | 33.2 | 1.2 |
| T07wt | Dong | 646 | 0.56 | 0.2 | 0.1 | 66.9 | 0.1 | 15.9 | 31.6 |
| T25wt | Tab.4 | 5154 | 5.01 | 17.4 | 4.8 | 90.0 | 1.6 | 17.3 | 3.3 |
| T10wt | Tab.4 | 1259 | 1.20 | 14.9 | 4.5 | 88.6 | 1.8 | 17.1 | 5.8 |
| T07wt | Tab.4 | 646 | 0.56 | 0.2 | 0.1 | 56.6 | 0.1 | 16.6 | 43.5 |

Values for each transcriptome (T25wt,T10wt,T07wt) are obtained from an average over three separate simulations. The tRNA column denotes the tRNA bias used in the simulation, either the tRNA abundances measured from Dong et al., or the inferred tRNA abundance based on the codon bias from the *E. Coli* ORFs (Table 4). Variables denote the following quantities: $N$—number of mRNAs in the transcriptome, $N_{nt}$—total number of nucleotides in the transcriptome in millions, $C_p$—average protein chain elongation rate (aa/sec), (%I,%E,%T, and %S)—Percentage of ribosomes (I)nitiating, (E)longating, (T)erminating, (S)talled, and %50S—Percentage of 50S ribosomal subunits that are free.

that it decodes for. It is important to note that, due to cross recognition of codons by multiple tRNAs, there are multiple solutions to this problem, and therefore a more optimal solution may exist.

As can be seen in Table 3, using the experimentally measured tRNA abundances from Dong et al. [26] results in extremely poor translation ($C_p \leq 6.0$ aa/sec) for all of the growth rates. However, by using a tRNA bias which matches the codon frequency in the ORFs of *E. Coli*, we can see that translation efficiency is dramatically improved for the T25wt and T10wt transcriptomes, while the T07wt transcriptome still suffers from poor transcription. Upon examination of the quasi-steady state distribution of tRNAs in free TC for the T07wt simulation, the tRNAs Arg4 and Arg5 which decode the ultra-rare arginine codons AGA and AGG have essentially zero tRNAs in free TC. This suggests that a minimal threshold of tRNAs may be important for ensuring efficient cycling through the recharging process, and additional constraints on tRNA abundances may exist.

Although using the tRNA abundance in Table 4 which matches the codon bias of the mRNAs improves translation in the wt transcriptomes, the translation dynamics do not match the experimental observations of Bremer and Dennis as well as the artificial transcriptome with tRNA abundances from Dong et al. [26]. There are many possible explanations for this, along with potential ways to tune the rates/concentrations to achieve a better fit to experiment for the wild-type transcriptomes. First, it may be that an alternative tRNA distribution which also matches the codon bias, is more efficient, or that the tRNAs need to slightly deviate from the codon bias for other unknown reasons. Second, it could be that the kinetic rates for the elongation process, which were optimized for the Dong tRNA abundances by Rudorf et al. [25] need re-calibrated, or other kinetic rates, such as those for aminoacyl-tRNA recharging, need altering. Third, there may be some known/unknown reactions for elongation or tRNA recharging that need to be accounted for in the model. Fourth, the concentrations of elongation factors such as Ef-Tu may be too low as increasing Ef-Tu increased the translational rates for T10a. Finally, there may be a combined tRNA and kinetic rate optimization which must be done in tandem, along with potentially other factors, to get a good fit of the wild-type transcriptomes to the experimental translational data of Bremer and Dennis [19].

## Discussion

In this work, I have developed a stochastic model of ribosome kinetics based on the [20] simulation algorithm which is capable of simulating translation on thousands of unique mRNAs

**Table 4. A theoretical estimate for the number of tRNA$^x$ in E. Coli. K12.**

| $x$ | $\mu = 0.70$ | $\mu = 1.07$ | $\mu = 2.50$ |
|---|---|---|---|
| Ala1 | 4718 | 9437 | 37748 |
| Ala2 | 1745 | 3491 | 13964 |
| Arg2 | 3177 | 6354 | 25416 |
| Arg3 | 366 | 733 | 2932 |
| Arg4 | 138 | 276 | 1104 |
| Arg5 | 76 | 153 | 612 |
| Asn | 2666 | 5333 | 21332 |
| Asp1 | 3488 | 6977 | 27908 |
| Cys | 792 | 1584 | 6336 |
| Gln1 | 1047 | 2094 | 8376 |
| Gln2 | 1972 | 3945 | 15780 |
| Glu2 | 3917 | 7834 | 31336 |
| Gly1 | 474 | 949 | 3796 |
| Gly2 | 815 | 1631 | 6524 |
| Gly3 | 3709 | 7418 | 29672 |
| His | 1541 | 3083 | 12332 |
| Ile1 | 3794 | 7588 | 30352 |
| Ile2 | 289 | 579 | 2316 |
| Leu1 | 1870 | 3740 | 14960 |
| Leu2 | 1509 | 3019 | 12076 |
| Leu3 | 2010 | 4020 | 16080 |
| Leu4 | 621 | 1243 | 4972 |
| Leu5 | 1251 | 2502 | 10008 |
| Lys | 2992 | 5985 | 23940 |
| Metm | 1701 | 3403 | 13612 |
| Phe | 2645 | 5290 | 21160 |
| Pro1 | 943 | 1887 | 7548 |
| Pro2 | 503 | 1007 | 4028 |
| Pro3 | 1567 | 3135 | 12540 |
| Ser1 | 1091 | 2182 | 8728 |
| Ser2 | 398 | 796 | 3184 |
| Ser3 | 1687 | 3374 | 13496 |
| Ser5 | 760 | 1520 | 6080 |
| Thr1 | 767 | 1535 | 6140 |
| Thr2 | 686 | 1373 | 5492 |
| Thr3 | 767 | 1535 | 6140 |
| Thr4 | 1440 | 2880 | 11520 |
| Trp | 1040 | 2080 | 8320 |
| Tyr1 | 964 | 1929 | 7716 |
| Tyr2 | 964 | 1929 | 7716 |
| Val1 | 2988 | 5977 | 23908 |
| Val2 | 1816 | 3632 | 14528 |

One possible solution for the number of tRNAs that would be observed in E. Coli K12 substrain MG1655 (accession code U00096) if it is assumed that tRNA abundances match the codon bias in the predicted ORFs.

with an explicit nucleotide sequence in an *in vivo* context. The model incorporates a number of important biophysical properties of the translational process simultaneously and accounts for different codon translation rates, which will in turn depend on tRNA abundance and usage by other mRNAs. Moreover, it accounts for the effects of individual ribosomes competing for the translation factors and a polysomal state of multiple ribosomes to be bound simultaneously on the mRNAs, while also taking into account the excluded volume of the ribosomes and the formation and resolution of traffic jams. The computational code is set up for the prediction of initiation rates on mRNAs based on sequence and secondary structure around the start codon, presenting an opportunity in the near future for experiment and theory to elucidate the impact of sequence on translation initiation.

As demonstrated above, my translational models predictions for ribosomal activity $\beta_r$ for a given growth rate and mRNA nucleotide content are consistent with experimental observations by [19] for *E. coli.* In addition, the increasing peptide chain elongation rates with increasing growth rates are consistent with experiment over at least three different growth rates [24]. However, unlike in [25], only a single set of kinetic parameters are needed for the elongation reactions in order to reproduce this phenomenon. Indeed, the translational kinetics model reported here displays classic features of a complex dynamical system, where properties emerge from the complex interactions of all the components. The peptide chain elongation rate is a clear example of this phenomenon, as increasing $C_p$ values emerge naturally as a result of the changes to the complex dynamics of all of the translational components that occur at increasing concentrations. Attempts to increase $C_p$ values by increasing the concentration of only one or a few components of the system had no effect, further illustrating the complexity of the translational process. This is in direct contrast to Vieira et al. [4], who have not seen this effect using a stochastic framework model of translation. This is likely due to the lack of binding reactions governing the formation of free TC as well as missing termination steps involving the recycling factors. This shows the importance for including these reactions in translational models.

However, one of the factors which seems to have a strong influence on the translational dynamics was the composition of the transcriptome, both in its codon bias relative to tRNA concentrations and its length and total nucleotide content. The ultra sensitivity of the translational process on the tRNA abundance relative to codon bias has also been noted by Rudorf et al. when Ef-Tu and TC formation reactions have been accounted for in their translational models [28, 30]. However, preliminary work on reproducing the observed translational dynamics with wild-type *E. Coli* transcriptomes using mRNAs from predicted ORFs showed that, while matching of the tRNA abundances to codon bias enhanced translational efficiency to values closer to what is expected experimentally, additional optimization of the parameters or the overall model is needed.

The implication of the results for tRNA abundances and their effects on translational efficency, as also discussed in Rudorf et al. [28], are that changes to the transcriptomes codon bias from over expression of particular mRNAs (such as lacz) due to cellular responses to the environment could potentially result in alterations to the elongation rates of other mRNAs due to redistribution of tRNAs in free TC. In cell free protein syntehsis applications, the results show a potential need for researchers to carefully tune tRNA concentrations to the demand implied by the transcriptomes codon usage to maximize protein synthesis. These observations points to a need to consider translational kinetics from a more holistic view, where the effects of translation due to the presence of a heterogeneous population mRNAs need to be considered. Moreover, the simulations with T10, T10a, T10b, and T10c, illustrate how ribosome density is likely controlled by a complex set of factors which include the overall number of ribosomes, mRNAs, as well as the lengths of the mRNAs and their relative abundances and ribosomal

initiation rates. As Bremer and Dennis have reported, observed ribosomal activity $\beta_r$ is roughly constant across the different growth rates of *E. coli* and the corresponding mRNA nucleotide content reported is approximately 1.0M, 2.0M, and 4.6M nucleotides for growth rates of $\mu$ = 0.7, 1.0, and 2.5 doublings per hour, respectively [19]. Although this is slightly different from the values of 0.56M, 1.2M and 5.0M nucleotides used here, the discrepancy is likely due to mRNAs having varying initiation rates and expression levels in reality, while here all mRNAs have been arbitrarily fixed to have maximal initiation rate. Taking into account different initiation rates for mRNAs will likely also result in the average distance between ribosomes being more in line with experimental measurements, which Bremer and Dennis compute to be roughly $d_r$ = 160 for $\mu$ = 1.0, compared with the value of $d_r$ = 93 I have computed here. However, adjustments to the transcriptome, and to some of the kinetic parameters for the initiation stage of translation, would require knowledge on the initiation rates for all mRNAs in the transcriptome, something beyond the scope of this initial work. Despite this, the translational model used here has elucidated a number of unexpected effects due to the composition of the transcriptome.

The model I developed opens up numerous possibilities for linking experimental measurements with theoretical predictions and for testing the physical feasibility of models that have been postulated based on experimental observations. Indeed, through testing and tuning of this model to recapitulate experimental observations, I found a number of experimentally hypothesised models were unable to result in a steady state of protein synthesis (see Supporting Information for discussion). Finally, it can also help in the development of simplified models through the choice of better model parameters. For instance, the observations from this more complex model can help guide TASEP models to consider additional features, such as dynamic initiation times, which may be critical for modeling specific features of the translational process.

## Methods

### Model construction and parametrization

Explicit details on the construction of the model, along with justification of the kinetic rates and concentrations of the translational factors, are based on a large background of experimental literature and full details can be found in the supporting information. Software and transcriptome data used in the simulations can be downloaded from http://www-users.york.ac.uk/ ∼ecd502/ or requested from the author via email.

### Construction of the transcriptome

The transcriptomes used in the simulations were constructed using the known *E. coli* proteome, which was obtained from the uniprot database https://www.uniprot.org/proteomes/ UP000000625. This database lists the number of amino acids in each of the 4353 identified proteins in the *E. coli* K12 strain, along with an amino acid sequence for the protein. In order to achieve a maximal translation rate of 22 codons per second at the highest growth rates, codon usage frequency in the mRNAs must match the abundances of the tRNAs which can decode them. Hence, the individual amino acid sequences were ignored and codons for the mRNAs were chosen according to the relative abundance of tRNAs in Table K in S1 Text, while preserving the reading frame length for each gene. For tRNAs such as tRNA-Lys, which is cognate to codons AAA and AAG, it is assumed that the number of tRNAs (4360 at $\mu$ = 1.0) implies an equal frequency usage of 2180 for both codons. Summing these contributions to the codons over all tRNAs and normalizing gives the frequency of codon usage, which is then sampled to construct representative mRNAs for each gene in the uniprot database.

To construct a transcriptome, individual mRNAs corresponding to a single gene are selected at random, and the number of copies for the mRNA is assigned into either a high (probability 5%), intermediate (probability 35%), or low (probability 60%) expression category. Copy numbers ($k$) for each mRNA are then determined by sampling from either a Poisson distribution, $p(k) = e^{-\lambda}\lambda^k/k!$, for the high expression ($\lambda = 6.8$), or a geometric distribution, $p(k) = (1 - \lambda)^k\lambda$, for the intermediate ($\lambda = 0.58$) and low ($\lambda = 0.93$) expression categories. Genes and their corresponding mRNA sequences are selected, one at a time, until a total nucleotide content and/or total number of mRNAs is achieved.

## Supporting information

**S1 Text. Supplementary information.**
(PDF)

## Acknowledgments

ECD wishes to thank Prof. R. Twarock and Dr. R. J. Bingham for careful reading of the manuscript and their helpful feedback.

## Author Contributions

**Conceptualization:** Eric Charles Dykeman.

**Data curation:** Eric Charles Dykeman.

**Formal analysis:** Eric Charles Dykeman.

**Funding acquisition:** Eric Charles Dykeman.

**Investigation:** Eric Charles Dykeman.

**Methodology:** Eric Charles Dykeman.

**Project administration:** Eric Charles Dykeman.

**Resources:** Eric Charles Dykeman.

**Software:** Eric Charles Dykeman.

**Validation:** Eric Charles Dykeman.

**Visualization:** Eric Charles Dykeman.

**Writing – original draft:** Eric Charles Dykeman.

**Writing – review & editing:** Eric Charles Dykeman.

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
