## [Decision Letter · Decision Letter 0]

29 Oct 2019

Dear Dr Dykeman,

Thank you very much for submitting your manuscript 'A stochastic model for simulating ribosome kinetics in vivo' for review by PLOS Computational Biology. Your manuscript has been fully evaluated by the PLOS Computational Biology editorial team and in this case also by independent peer reviewers. The reviewers appreciated the attention to an important problem, but raised some substantial concerns about the manuscript as it currently stands. While your manuscript cannot be accepted in its present form, we are willing to consider a revised version in which the issues raised by the reviewers have been adequately addressed. We cannot, of course, promise publication at that time.

Sincerely,

Attila Csikász-Nagy

Associate Editor

PLOS Computational Biology

William Noble

Deputy Editor

PLOS Computational Biology

[LINK]

Reviewer's Responses to Questions

**Comments to the Authors:**

Reviewer #1: The manuscript develops a stochastic model of protein synthesis, which is used to simulate translation in E. coli cells. I have problems fully evaluating this manuscript, due in part to the fact that the important aspects of the work seem incompletely described, and that novelty is claimed in a way that I do not completely agree with.

1) The introduction gives the impression that previous work on modelling protein synthesis has relied almost exclusively on mathematical modelling via the TASEP, that very little work was done on stochastic modelling, and that none of the latter attempted genome-wide models of translation. This is factually untrue – a detailed Gillespie-based model of translation was published eg in 2007 (PMID 17897886) and this was used for modelling genome-wide translation in yeast (22965119, 23791185). The author even cites a more recent model as using a similar approach – simulating translation in the cell or in a cell-free extract into fundamentally different apart from the parameterisation. The principal findings in the publications cited above on translation in vivo do not seem fundamentally different from those described here.

2) Although I see strong overlap with previous work, there is some novelty in this manuscript but this is not clearly described. First, the detailed model for translation initiation is novel as far as I know, although this is qualified in the text as “somewhat hypothetical”. If there is uncertainty over the structure or parameterisation of this part of the model, this should have been discussed in more detail. Second, the technical approach used for the modelling described on page 4 seems interesting, but is described without any validation that this produces the same results as the original approach described by Gillespie, or as the faster derivatisations of this approach like tau-leap etc. In my view a more efficient algorithm for stochastic modelling of chemical systems would be very useful, but this would need to be validated by comparing results with more established approaches. I am assuming that the approach used here is more efficient than existing stochastic modelling approaches, if not, it should be explained why a new approach was necessary. It should also be explained how spatial information (ribosome positioning on the mRNAs) was encoded, as this is the biggest problem in standard ODE-based approaches and the reason why papers such as the one cited in the text require such high numbers of equations even for short mRNAs.

3) The approach for constructing the transcriptome appears so nonsensical that I wonder whether I misunderstand something fundamental. The author appears to say that they constructed the main transcriptome used in the simulations from proteome data, in such a way that the codon usage of the transcriptome fits the assumed tRNA pool better than experimentally reported transcriptomes. If I understood this correctly, they would model an entirely unphysiological transcriptome?

Reviewer #2: In his manuscript, Dr. Dykeman presents to my knowledge the first computational model of in-vivo protein synthesis that is built on a close-to complete set of reactions and captures the stochastic nature of this highly complex process. He uses this model to study fundamental aspects of translation, such as the effect of the system’s tRNA and codon compositions on the availability of free ternary complexes.

I find this work highly interesting and believe that it significantly advances the field. In particular, it is the first computational model of protein synthesis for a broad range of bacterial growth rates that uses only a single set of kinetic parameters. A convincing validation of the approach is given by comparing the numerical results with experimental observations.

The manuscript is very well organized and the text is very well written, clear and easy to follow. An exceptional comprehensive, thorough and transparent discussion about the limits of the model, which covers many challenges that are common for this type of modeling (e.g. a consistent parameterization), is included.

There is only one minor issue I kindly ask the author to comment on:

In the abstract you mention that “several biological phenomena are predicted as emergent properties of this complex translational dynamics”. Could you please be a bit more specific so that the readers know what to expect from your paper?

Reviewer #3: The paper of Dykeman deals with the simulation of in vivo translation kinetics in E. coli outlining that the cellular context is issential to understand experimental findings.

While the approach is basically highly attractive and deserves publication, the paper has some shortcomings that need to be eliminated before publication:

1) The author outlines the in vivo character of the simulation. However, the reader cannot understand how (and why) the arbitrary transcriptomes have been designed (lines 154 -156). There are numerous E. coli publications covering genome-scale transcript data. Why should the transcript artifact of the author be superior to experimental measurements?

2) The adaptation of nucleotide sequence - apparently a crucial step for sequenced based modelling - is not properly explained (lines 158 - 159). Actually, the need to do so does not get clear at all.

3) Apparently, initiation modelling has been simplified ignoring the formation of S30-IF1 and S30-IF2 - why?

4) In general: Considering the 'artificial' transcriptome for simulating translation is certainly important and highly attractive. Nevertheless, some findings were already published by other others who focused on single genes alone. Examples are: Niess et al. already discussed impacts on tRNA shortage and stochasticity on translation efficiency, Vieira et al implemented stochastic translational elements, Arnold et al. already published a very complex ODE model in 2005. The author is invited to check further details in the following list:

Arnold, S., Siemann-Herzberg, M., Schmid, J., and Reuss, M. (2005). Model-based Inference of Gene Expression Dynamics from Sequence Information. In Biotechnology for the Future, (Springer, Berlin, Heidelberg), pp. 89–179

Mehra, A., and Hatzimanikatis, V. (2006). An Algorithmic Framework for Genome-Wide Modeling and Analysis of Translation Networks. Biophysical Journal 90, 1136–1146.

Nieß, A., Failmezger, J., Kuschel, M., Siemann-Herzberg, M., and Takors, R. (2017). Experimentally Validated Model Enables Debottlenecking of in Vitro Protein Synthesis and Identifies a Control Shift under in Vivo Conditions. ACS Synth. Biol. 6, 1913–1921.

Nieß,A., Siemann-Herzberg,M. and Takors,R. (2019) Protein production in Escherichia coli is guided by the trade‑off between intracellular substrate availability and energy cost. Microbial Cell Factories, 18

Shaw, L.B., Zia, R.K.P., and Lee, K.H. (2003). Totally asymmetric exclusion process with extended objects: A model for protein synthesis. Phys. Rev. E 68, 021910.

Vieira, J.P., Racle, J., and Hatzimanikatis, V. (2016). Analysis of Translation Elongation Dynamics in the Context of an Escherichia coli Cell. Biophys. J. 110, 2120–2131.

Zouridis, H., and Hatzimanikatis, V. (2007). A Model for Protein Translation: Polysome Self-Organization Leads to Maximum Protein Synthesis Rates. Biophysical Journal 92, 717–730.

Zouridis, H., and Hatzimanikatis, V. (2008). Effects of Codon Distributions and tRNA Competition on Protein Translation. Biophysical Journal 95, 1018–1033.

**Have all data underlying the figures and results presented in the manuscript been provided?**

Reviewer #1: No: I could not locate data underlying figures.

Reviewer #2: Yes

Reviewer #3: Yes

PLOS authors have the option to publish the peer review history of their article (what does this mean?). If published, this will include your full peer review and any attached files.

Reviewer #1: Yes: Tobias von der Haar

Reviewer #2: No

Reviewer #3: No

---

## [Decision Letter · Decision Letter 1]

24 Nov 2019

Dear Dr Dykeman,

Thank you very much for submitting your manuscript 'A stochastic model for simulating ribosome kinetics in vivo' for review by PLOS Computational Biology. Your manuscript has been fully evaluated by the PLOS Computational Biology editorial team and in this case also by independent peer reviewers. The reviewers appreciated the attention to an important problem, but raised some substantial concerns about the manuscript as it currently stands. While your manuscript cannot be accepted in its present form, we are willing to consider a revised version in which the issues raised by the reviewers have been adequately addressed. We cannot, of course, promise publication at that time.

Sincerely,

Attila Csikász-Nagy

Associate Editor

PLOS Computational Biology

William Noble

Deputy Editor

PLOS Computational Biology

[LINK]

Reviewer's Responses to Questions

**Comments to the Authors:**

Reviewer #1: I thank Dr Dykeman for the detailed response to my comments.

I am generally satisfied with the response to the first two comments. With respect to the “novelty” aspect, my evaluation was too focused on the translation process itself and I had not considered all the additional reactions included in the model, so upon reflection I agree with Dr Dykeman’s argument in this respect. The details of the modified Gillespie algorithm had escaped me but are now clearer and easier to understand for non-experts such as myself.

I do however have continuing concerns about the decision to base the majority of the study on a non-physiological transcriptome. Expressed in a slightly snide way to drive home the point, the message is essentially that “because my model does not fit well with experimental data, I make up a dataset that fits the model better and base my study on that”. I thought about this for quite a while because I would like to be fair to the study and its author, but in the end this is not an approach I wish to condone for the following reasons:

1) The assumption that it is the experimental data that are are inaccurate, rather than the model, seems arbitrary in the first instance.

2) Even if the model is correct and some of the experimental data inaccurate, the transcriptome replacement strategy is underpinned by the very specific but arbitrary notion that it must be the tRNA data that are wrong. Why can it not be the target amino acids per second data that are wrong, or the recharging data (nb “recharging” in the manuscript should actually be “nucleotide exchange” – recharging traditionally refers to tRNA synthetase reactions). It is clearly acknowledged that altering elongation factor re-charging can produce high elongation rate with the actual E. coli transcriptome...

3) One of the motivations for replacing the transcriptome is the assumption that tRNAs and codon usage should match perfectly (most directly expressed in lines 283-287). This seems to take that literature observation that there is some correlation between codon usage and tRNA content and interpret it very strongly.

4) Overall, I think this study could be re-framed as one that highlights apparent mismatches in experimental data revealed by the modelling, and carefully discusses the potential implications. However, I do not think the current framing where an (in my view) uncertain work-around to the mismatch is provided and this is then used to make strong statements about biological implications works very well.

Reviewer #2: Review of PCOMPBIOL-D-19-01692R1

My original comment was fully addressed and I still believe that this manuscript deserves publication.

However, I now disagree with some parts of the revised manuscript and these issues should be addressed prior to publication.

In my opinion, the great advancement of the presented work is the following:

by integration of many different biomolecular processes that govern protein biosynthesis into a “holistic” model, many important features of translation (growth rate dependence of translation rates, sensitivity to codon/tRNA composition, usage of the machinery) are shown to be inherent to the system and can be explained without any further assumptions. Neglecting some parts of the system, such as ternary complex formation, leads to a loss of these characteristic features.

I agree that it is important to stress this point but while doing so the author must be careful to not make the impression as if some of the _individual features_ have not been studied before. In particular, in the new text the author writes

- “While previous stocahstic models have examined competiton between tRNAs in TC for the A-site in elongating ribosomes [2–4], they have not considered the recharging reactions, specifially GTP/GDP exchange on Ef-Tu and formation of TC” (l 324ff)

- “…the model I report on here takes into account effects from the additional competition that occurs between aminoacyl-tRNAs for Ef-Tu:GTP during formation of free TC after recharging of tRNA by aminoacyl-TRNA synthase” (l 422ff)

- “… the effects on the translational rate from the competition between aminoacyl-tRNAs for formation of free TC is much more dramatic” (l 426f)

It is not true that the great influence of ternary complex formation on the translation rate has not been described before. In contrast, that the competition for EF-Tu during TC formation has a tremendous effect on the overall rate of protein synthesis was recently studied and described in great detail (PMID 31369559). Note that neglecting the GDP/GTP exchange does not cause a fundamental difference between the published and your model, because including the GDP/GTP exchange basically reduces the effective concentration of EF-Tu available for ternary complex formation. The mentioned paper demonstrates that such a reduction (independent of its origin) strongly affects protein synthesis.

- “Thus, situations where the relative abundances of tRNAs do not match their demand implied by the codon bias of the transcriptome will result in shifts in the amount of tRNA available in free TC” (l 330ff)

- “This observation implies that tRNA abundances must closely match the levels of demand placed on them as a result of the codon bias in the transcriptome for efficient translation to occur” (l 428ff)

In the paper mentioned above (PMID 31369559) it was shown in detail that a disbalance between codon usage and tRNA abundance is the driving force behind the sensitive dependence of translation on TC formation (i.e., EF-Tu availability). That this causes “shifts in the amount of tRNA available in free TC” was also already demonstrated previously (PMID 26270805).

- “… removing the reactions involved in Ef-Ts mediated GDP/GTP exchange on Ef-Tu, a feature that is often neglected in translational models, resulted in similar average translational speeds for T10a and T10 transcriptomes, despite the difference between the demand for tRNA species implied by the codon usage in T10a and the tRNA abundances present”(l 430ff)

Again: That (effectively) increasing the amount of EF-Tu available for TC formation (which is exactly what happens when GDP/GTP exchange is ignored) relieves the sensitivity of the speed of translation for disbalances in the tRNA/codon composition was demonstrated before (PMID 31369559). The observation made by the author is thus in full agreement with the previously published model and the statement is important to make because in most translation models the strong impact of the availability of EF-Tu for TC formation is indeed ignored. But again, the author should mention that he is not the first/only to make this observation and must put his results in the context of previous publications.

Side note: This implies that the strong dependency of the results on the chosen transcriptome is “caused” by the chosen value for the EF-Tu concentration. You could check if the differences between T10 and T10a still remain when you increase the EF-Tu by e.g. 10, 20 or 50 % (but keeping the GDP/GTP exchange reaction). At some point the differences should vanish.

In this context one could also discuss that EF-Tu has many functions in bacteria in addition to its main role as elongation factor. That means that other cell processes sequester EF-Tu, thus reduce the amount of EF-Tu available for TC formation and increase the need for a well-balanced tRNA/codon composition.

Reviewer #3: The author has covered most questions of the reviewer very well. Some optimization potential still exists with respect to the cited literature. But this is a very minor remark.

**Have all data underlying the figures and results presented in the manuscript been provided?**

Reviewer #1: Yes

Reviewer #2: Yes

Reviewer #3: Yes

PLOS authors have the option to publish the peer review history of their article (what does this mean?). If published, this will include your full peer review and any attached files.

Reviewer #1: Yes: Tobias von der Haar

Reviewer #2: No

Reviewer #3: No

---

## [Decision Letter · Decision Letter 2]

19 Dec 2019

Dear Dr Dykeman,

We are pleased to inform you that your manuscript 'A stochastic model for simulating ribosome kinetics in vivo' has been provisionally accepted for publication in PLOS Computational Biology.

In the meantime, please log into Editorial Manager at https://www.editorialmanager.com/pcompbiol/, click the "Update My Information" link at the top of the page, and update your user information to ensure an efficient production and billing process.

One of the goals of PLOS is to make science accessible to educators and the public. PLOS staff issue occasional press releases and make early versions of PLOS Computational Biology articles available to science writers and journalists. PLOS staff also collaborate with Communication and Public Information Offices and would be happy to work with the relevant people at your institution or funding agency. If your institution or funding agency is interested in promoting your findings, please ask them to coordinate their releases with PLOS (contact ploscompbiol@plos.org).

Thank you again for supporting Open Access publishing. We look forward to publishing your paper in PLOS Computational Biology.

Sincerely,

Attila Csikász-Nagy

Associate Editor

PLOS Computational Biology

William Noble

Deputy Editor

PLOS Computational Biology

Reviewer's Responses to Questions

**Comments to the Authors:**

Reviewer #1: Once again I thank Dr Dykeman for the thorough and engaged response. The arguments put forward in his response make sense, and I note that the rebuttal is essentially framed as the discussion I had suggested of whether the alternative transcriptome is useful (and what for). I would have preferred if more of this discussion had found its way into the actual manuscript. However, given that the other two reviewers did not share my concerns over the use of the alternative transcriptome, and that I think the manuscript as it stands sufficiently alerts readers to the implications of using this transcriptome, I'm happy to not insist further on this issue.

Reviewer #2: All of my concerns have been resolved and I have no further comments.

**Have all data underlying the figures and results presented in the manuscript been provided?**

Reviewer #1: Yes

Reviewer #2: Yes

PLOS authors have the option to publish the peer review history of their article (what does this mean?). If published, this will include your full peer review and any attached files.

Reviewer #1: Yes: Tobias von der Haar

Reviewer #2: No

---

## [Editor Report · Acceptance letter]

22 Jan 2020

PCOMPBIOL-D-19-01692R2 

A stochastic model for simulating ribosome kinetics in vivo

Dear Dr Dykeman,

I am pleased to inform you that your manuscript has been formally accepted for publication in PLOS Computational Biology. Your manuscript is now with our production department and you will be notified of the publication date in due course.

With kind regards,

Laura Mallard
